# How to Represent Paintings: A Painting Classification Using Artistic Comments

**DOI:** 10.3390/s21061940

**Published:** 2021-03-10

**Authors:** Wentao Zhao, Dalin Zhou, Xinguo Qiu, Wei Jiang

**Affiliations:** 1College of Mechanical Engineering, Zhejiang University of Technology, Hangzhou 310023, China; zhaowentao@zime.edu.cn (W.Z.); weij@zjut.edu.cn (W.J.); 2School of Intelligent Transportation, Zhejiang Institute of Mechanical & Electrical Engineering, Hangzhou 310053, China; 3School of Computing, University of Portsmouth, Portsmouth PO1 3HE, UK; dalin.zhou@port.ac.uk

**Keywords:** graph convolutional networks, art classification, machine learning, natural language processing

## Abstract

The goal of large-scale automatic paintings analysis is to classify and retrieve images using machine learning techniques. The traditional methods use computer vision techniques on paintings to enable computers to represent the art content. In this work, we propose using a graph convolutional network and artistic comments rather than the painting color to classify type, school, timeframe and author of the paintings by implementing natural language processing (NLP) techniques. First, we build a single artistic comment graph based on co-occurrence relations and document word relations and then train an art graph convolutional network (ArtGCN) on the entire corpus. The nodes, which include the words and documents in the topological graph are initialized using a one-hot representation; then, the embeddings are learned jointly for both words and documents, supervised by the known-class training labels of the paintings. Through extensive experiments on different classification tasks using different input sources, we demonstrate that the proposed methods achieve state-of-art performance. In addition, ArtGCN can learn word and painting embeddings, and we find that they have a major role in describing the labels and retrieval paintings, respectively.

## 1. Introduction

Due to the large-scale digitization of art works over the past two decades, art coupled with computer technology has become a research hotspot. Most online painting collections include various metadata, often written by art experts, to provide context and improve audience understanding. For example, WikiArt (http://www.wikiart.org, accessed on 25 December 2020) includes artist, style, genre, media, date, dimensions and even artistic comments describing the paintings. Scholars who have spent years analyzing and learning the specifics and nuances of fine art can easily identify metadata associated with a painting [1]; however, identifying metadata is difficult for general audiences who lack expertise. As deep learning has developed, more works have focused on using learned features to conduct art classification [2,3,4,5,6,7]. In other words, by training computers using paintings previously labeled by human experts, the machines can learn the image features and classify labels for the images automatically. To the best of our knowledge, the features used thus far in art classification tasks have been extracted from the images themselves. Similar to humans, researchers have used paintings to teach machine knowledge about the arts in most previous works [8,9,10,11,12,13,14].

Language has a referential function, i.e., it can refer to objects in the world, and the object also has a visual representation; therefore, language can (at least to some extent) describe visual perception. Inspired by this phenomenon, we use the artistic comments associated with paintings to perform artwork classification. This approach shifts the problem from one of computer vision to one involving natural language processing (NLP).

To summarize, our contributions are as follows:We propose a novel graph neural network method for art classification that uses features extracted from textual comments about art. In addition, on this basis, we propose a multitask learning model (MTL) to solve the different tasks using one model. This approach encourages the models to find common elements (and hence the context) between the different tasks. A comparison shows that our proposed approach performs state-of-the-art compared with traditional benchmarks that apply vision-based and text-based methods to classify art.Our method performs word embedding and creates labels using the dimension that has the highest value. The mean of the label is the same as the painting label; thus, by extracting the top 10 words that had higher values in each category, we find that the extracted words are highly correlated descriptions of labels.We analyze the SemArt dataset, including the class distribution. We create visualizations to analyze the results of the art classification for different tasks using ResNet50 and ArtGCN models. Overall, the classification effect improves when using ArtGCN—but not for all specific categories. These 2 methods include using the pixels of the painting itself (ResNet50) and art comments (ArtGCN). Here, we compare two ways of representing paintings using neural networks.Based on the trained classification models, we develop a painting retrieval system and find that both methods achieve good performance. An analysis of the retrieval results further illuminates the differences in how the 2 models with different input sources work.

## 2. Related Work

### 2.1. Art Classification

Early works concentrating on automatic art analysis first extracted hand-crafted features from images and then performed classification using traditional machine learning methods [15,16,17] developed the QArt-learn approach to categorize paintings into art styles. The model was implemented using k-nearest neighbor and support vector machine (SVM) algorithms with QCD color features. In recent years, deep CNNs have achieved many successes on computer vision tasks using large hand-labeled datasets, such as the ImageNet dataset [18]. Recent studies to solve classification problems have been dominated by CNNs [11,19,20,21].

### 2.2. Text Classification

Text classification is a fundamental task in natural language processing (NLP). The most representative deep networks are sequential-based learning models, including both CNNs [22,23,24] and recurrent neural network (RNN) models [25,26,27]. The developed models are effective and widely used, but due to the model architectures, they focus primarily on text features extracted from local consecutive word sequences. Since 2018, Transformers have been the state-of-the-art architecture for most NLP tasks. The current SoTA approach to basically text classification is taking a pretrained Transformer [28] model (such as BERT [29]) and finetunes it for a particular task.

### 2.3. Graph Neural Networks

Recently, graph neural networks have received increasing attention in many applications [30,31,32,33,34]. Kipf and Welling presented a simplified graph neural network model called a graph convolutional network (GCN) [35]. By exploiting message passing (or equivalently, various neighborhood functions), GCNs have achieved high scores on several benchmark graph datasets. Recently, graph neural network (GNN) models have also attracted widespread attention and have been successfully applied to solve text classification problems by using global word co-occurrence information in a corpus [36,37,38].

## 3. Datasets

To achieve our goal of classifying paintings using artistic comments, we adopted the SemArt dataset collected from [39] to classify various metadata, including type, school, timeframe and author. In recent years, several datasets have been used to classify artworks, such as Painting91 [40], WikiArt-WikiPaintings [41] and MultitaskPainting100k [42]; however, these are labeled only by artists, styles and genres. Previous works have focused on the task of identifying paintings based on different artist representations. However, representing paintings is not an easy task, and paintings typically contain no more than 3 metadata points. In contrast, the paintings in the SemArt dataset include 8 types of labeled metadata, including artistic comments, authors, titles, techniques and formats, data regarding when the painting was created, types, schools and timeframes. These metadata improve the flexibility with which computers can represent paintings. Some examples of the dataset records are shown in Figure 1. For space reasons, we simplified artistic comments in the right-hand examples.

### 3.1. Data Analysis

The types used in our paper are as follows: religious, portrait, landscape, mythological, genre, still-life, historical, other, interior and study. The schools are grouped into 24 classes, e.g., Italian, English, American, etc. If the paintings are not labeled as belonging to any of these styles, we label them “unknown.“ Only 13 paintings are labeled unknown, indicating that these paintings do not belong to any of the 24 classes. We create timeframes consisting of every 50 years between 1050 and 1900 to determine the approximate date a painting was created. There are a total of 17 timeframes. In addition, 17 paintings in the dataset were not created between 1050 and 1900. Clearly identifying the authors of the paintings in the dataset is difficult because there are so many possible authors in the datasets. To better distinguish among authors, we selected 349 authors and labeled the others as unknown. For example, the artist Vincent van Gogh is the author of 327 paintings. A total of 8257 paintings are labeled as having an unknown author. In total, the dataset includes 11 styles, 25 classes, 18 timeframes and 350 authors (including the unknown labels). The distributions of these attributes can be seen in Figure 2. It can be found that class distribution is uneven, which makes sorting difficult. Figure 3 shows that the painting skill of every artist can vary widely. For example, all the paintings from Giotto di Bondone are religious, while Vincent van Gogh and Rembrandt Harmenszoon van Rijn are more diverse. Timeframes and schools are also highly relevant to author classification. For example, the paintings of the artist Giotto di Bondone belong to Italian (school) and the 1301–1350 timeframe, while the paintings of Rembrandt Harmenszoon van Rijn belong to the Dutch school and the 1601–1650 timeframe. In addition, paintings from Vincent van Gogh, belong to Dutch school and the 1851–1900 timeframe. The dataset is randomly divided into training, validation and test sets with 19,244, 1069, and 1069 images in each set, respectively. All the data are summarized in Table 1. The meanings of words, nodes and average length will be explained in Section 4.

## 4. Method

We first collected all the artistic comments including train, validation and test splits associated with the paintings and processed the texts. In detail, we cleaned the texts using preprocessing techniques such as changing case, correcting split characters (as in [22]), removing stopwords defined in NLTK and low frequency words appearing less than 5 times. After this process, the average length of an artistic comment decreased to 59.27 words, and the total number of unique words was 17,944, as shown in Table 1. Then, we build a large and heterogeneous text graph G=(V,E), where *V* and *E* denote the sets of nodes and edges in the graph, respectively. There are nodes of two types:Nodes representing artistic comments represented as TF-IDF weighted bag of words.Nodes that correspond to unique words.

The number of nodes |V| is equal to the sum of the number of artistic comments and unique words in the corpus. We used a one-hot vector as the input features of the nodes, which means that an input feature *X* is an identity matrix of |V| dimensions. The weights of the edges between the documents and words were calculated using the term frequency-inverse document frequency (TF-IDF):(1)$$TF-IDF_{i,j}=TF_{i,j}\times IDF_{i}$$
(2)$$TF_{i,j}=\frac{n_{i,j}}{\sum_{k}n_{k,j}}$$
(3)$$IDF_{i}=\frac{\left|D\right|}{1+\left|j:t_{i}\in d_{j}\right|}$$
where $n_{i,j}$ denotes the number of times word $t_{i}$ appears in an artistic comment, and $TF_{i,j}$ denotes the frequency of that word in an artistic comment. The term |D| represents the total number of artistic comments, and $\left|j:t_{i}\in d_{j}\right|$ denotes the number of artistic comments that contain entry $t_{i}$. The value of the TF-IDF index is higher when a word appears frequently in an article but appears less frequently among all documents. We employ pointwise mutual information (PMI) to calculate the weights of the edges between 2 word nodes. To calculate the results, we first applied a fixed-size sliding window to all the artistic comments to gather co-occurrence statistics:(4)$$\mathrm{PMI}\left(i,j\right)=\log\frac{p(i,j)}{p(i)p(j)}$$
(5)$$p\left(i,j\right)=\frac{W(i,j)}{W}$$
(6)$$p\left(i\right)=\frac{W(i)}{W}$$
where W(i,j) is the number of sliding windows containing both the words *i* and *j*, while W(i) is the number of sliding windows that contain word *i*. *W* is the total number of sliding windows. We adopt only the positive values that imply high semantic correlations between the words in a corpus. Formally, the weight of edge between nodes *i* and *j* is defined as:(7)$$A_{ij}=\left\{\begin{array}{cc} \mathrm{PMI}(i,j) & i,j\;\mathrm{are}\;\mathrm{words},\;\mathrm{PMI}(i,j)>0\\ {\mathrm{TF-IDF}}_{ij} & i\;\mathrm{is}\;\mathrm{artistic}\;\mathrm{comment},\;j\;\mathrm{is}\;\mathrm{word}\;\\ 0 & \;\mathrm{otherwise}\;. \end{array}\right.$$

After building the graph, we use a two-layer GCN model to perform the art classification task. A GCN [35] can operate directly on a graph and propagates features between neighborhoods. Mathematically, GCN follows this formula:(8)$$h^{\left(l+1\right)}=\sigma \left({\tilde{D}}^{-\frac{1}{2}}\tilde{A}{\tilde{D}}^{-\frac{1}{2}}h^{\left(l\right)}W^{\left(l\right)}\right)$$
where $h^{\left(l\right)}$ denotes the $l^{\left(th\right)}$ layer in the network, σ is the nonlinearity, and *W* is the weight matrix for this layer. ${\tilde{D}}^{-\frac{1}{2}}\tilde{A}{\tilde{D}}^{-\frac{1}{2}}$ indicates a renormalization trick in which a self-connection appears for each graph node. Therefore, $\tilde{D}$ is the corresponding degree matrix of A+I, and $\tilde{A}=A+I$. The shape of $H^{\left(0\right)}$ is N×D, where *N* is the number of nodes and *D* is the number of input features. As an alternative method of understanding, the following formula can also be used to describe the models:(9)$$h_{i}^{\left(l+1\right)}=\sigma \left(\sum_{j\in N(i)\cup \{i\}}\frac{1}{\sqrt{\deg(i)}\cdot \sqrt{\deg(j)}}\cdot \left(h_{j}^{\left(l\right)}W^{\left(l\right)}\right)\right)$$
where $\sqrt{\deg(i)}$ and $\sqrt{\deg(j)}$ are the degrees of nodes *i* and *j*, respectively, N(i) is a neighbor of node *i*, and σ is the activation function; here, we used ReLU as the activation function. In our experiments, we found that using a deeper model can lead to oversmoothing and the gradient explosion problem [43,44,45]. In this case, two layers are sufficient to allow messages to pass among nodes:(10)$$X=L^{sym}\mathrm{ReLU}\left(L^{sym}XW_{0}\right)W_{1}$$
where $L^{sym}$ is a symmetric normalized Laplacian, $L^{sym}={\tilde{D}}^{-\frac{1}{2}}\tilde{A}{\tilde{D}}^{-\frac{1}{2}}$. The meanings of ${\tilde{D}}^{-\frac{1}{2}}$ and $\tilde{A}$ are introduced in Equation (Equation 8). $W_{0}\in R^{N\times D}$, $W_{1}\in R^{D\times class}$. Here, *N* is the number of nodes, and *D* is the embedding size of the middle layer. class denotes the number of labels. The loss function is defined using the cross-entropy loss function.
(11)$$\begin{array}{cc} \mathrm{loss}(x,\;\mathrm{class}\;) & =-\log\left(\frac{\exp(x[\mathrm{class}])}{\sum_{j}\exp\left(x\left[j\right]\right)}\right)\\ \; & =-x\left[\mathrm{class}\right]+\log\left(\sum_{j}\exp\left(x\left[j\right]\right)\right) \end{array}$$
where *x* and class are the output training nodes of the model and the targeted label of the training nodes being learned, respectively. We first process the data and build the graph; then we set up an early termination mechanism: training stops when loss_val does not decrease for lr_patience or when the maximum number of epochs is reached. For the mtl-ArtGCN, we use four different GCN layers in the second layer of the model to satisfy different tasks, and then calculate the loss using the outputs, the weight of each loss is set to 0.25. We used this trained model to calculate the classification accuracy. There is no need to interface for the test samples because the value and test node information is also updated during the training. When training is complete, we extract the embedding of the value and test nodes and use the dimension in the node information where the vector value is the largest as the predict label. Then, we calculate the validation and test accuracies using the true and predict labels.

## 5. Experiments

### 5.1. Hyperparameter Selection

To select the appropriate hyperparameters, we first set the learning rate to 0.02, dropout to 0, and $L_{2}$ loss weight to 0. We set the maximum number of epochs to 200, lr_patience to 10 and adopted the Adam [46] optimizer to learn the parameters. We tested different hyperparameter combinations via a grid search to study the parameter sensitivity of the models. All the experiments were executed 4 times to obtain the mean±std. Figure 4 shows the test accuracy under different hyperparameters, including the sliding window sizes and the mid-layer dimensions. The first row shows the test accuracy under different sliding window sizes and reveals that the test accuracy first increases as the window size increases but no longer changes much beyond a window size of 20. These results show that adopting a word sliding window size that is too small generates insufficient word occurrence information, while a window that is too large may introduce some invalid edges. The second row shows the test accuracy under different mid-layer dimensions. A too-low mid-layer embedding dimension does not propagate node label information well, while a too-high value does not improve the classification performance and time consumption. After tuning the hyperparameters, we used a mid-layer dimension of 300 and a window size of 20 for the remaining experiments.

### 5.2. Baselines

We compared our models with the results of computer vision learning methods using ResNet50, ResNet101 and ResNet152 [47], which have achieved excellent image recognition performance. To adapt the models for art classification, we modified the last fully connected layer to match the number of classes in each task. For the mtl-resnet50, we replace the last fully connected layer with four different fully connected layers to classify the paintings in different tasks in one model, the loss weight calculated by these different fully connected layers is set to 0.25 for all the tasks. For the kgm-resnet50 [48], the author use the contextual information which is obtained from capturing relationships in an artistic knowledge graph built with non-visual artistic metadata to help train the model. Due to the limited number of images, we used transfer learning, which the model parameters are first trained on ImageNet and then fine-tuned for art classification. In addition, we compare ArtGCN with multiple state-of-the-art text classification methods as follows: (1) TF-IDF + LR, which we have described in Section 4 and logistic regression is used as the classifier; (2) fastText [49], which treats the average of word or n-gram embeddings as document embeddings and a linear classifier to train the model. We evaluate the model with and without bigrams; (3) fine-tuned RoBERTa [50], which we fine-tuned the pretrained RoBERTa models in the SemArt. In general, we focus on visual-based and text-based painting classification efforts. The classification efforts of two different domain classification models for the same task are compared.

## 6. Results and Discussion

### 6.1. Results

Table 2 shows the results of the art classification. MTL denotes the multitask learning model [48], which is trained to learn multiple artistic tasks jointly. The results show that various classification tasks (including type, school, timeframe and author) perform better when using ArtGCN. Compared with the ResNet50 results, ArtGCN increased the type, school, timeframe and author classification accuracies by 5.00%, 23.90%, 21.11%, and 26.03%, respectively. These results indicate that our method can achieve huge improvements, especially for the school, timeframe and author tasks, and demonstrate the effectiveness of using art comments to represent art. By comparing with the previous state-of-the-art text classification methods, our method still achieved good performance, especially in timeframe and author classification. The parameters of popular RoBERTa model have 124.66 M, while our model only have 11.80 M. It shows our model has also greatly improved in terms of the number of parameters. We also find the RoBERTa and fastText methods, which text-based in artist classification is less effective than visual-based methods. Therefore, text-based methods are not always better than visual-based methods and a well-designed algorithm is also important to obtain state-of-the-art accuracy. Compared with the MTL-ArtGCN results, the classification results differ only slightly, which shows that different tasks involved in art classification have substantial commonalities. In other words, the first layer of the model learns the common parameters for different tasks. This is consistent with our common sense. Determining the artist of a painting can help in determining the nationality (school) and period (timeframe) of the painting and has some relevance to the style.

### 6.2. Word Embedding

The main working mechanism of a GCN is the propagation of node features through neighborhood nodes, through which the model achieves better node classification performance. As described in Section 4, we create a graph using ArtGCN in which the nodes are composed of both artistic comments and words. Only the labeled nodes form the training set of the artistic comments, and the models cannot identify which category the nodes belong to artistic comments or simply words. Therefore, we conduced some experiments. For the word node embeddings, we first performed normalization using the SoftMax function and considered the dimension with the highest value as a word label in the output embedding of the model, similar to the model prediction of the artistic comment node categories. The dimensions of the word labels are equal to the number of painting classes. We set the word labels as the image classes, e.g., if a word label is 0 in the style classification task, and a label of 0 for the painting represents religion; then the word belongs to the religion label. Figure 5 shows a t-SNE [51] visualization of the word embeddings. It can be found that words with the same label are close together, which means that most words are closely related to the painting classes.

After the embedding normalized by the SoftMax function, we show the top 10 words with the highest values in specific dimensions based on labels in Table 3 and find that the words are highly associated with the descriptions of the labels. For example, the religious words ‘saints’, ‘triptych’, and ’mary’ in column 1 express religion well, while words in other columns that show similar phenomena (e.g., estuary, views, and coastal) are highly related to landscapes. This means that the top 10 words with the highest values associated with each class are interpretable and can be used to describe the classes.

Table 4 shows the results for the school classification tasks. The output of the models has 25 dimensions (equal to the number of class labels). The calculation method used is the same as that for the style tasks, and we reach the same conclusions as for the style tasks. When talking about Italian, we are always reminded of Gian Giacomo Poldi Pezzoli, the founder of the Museo Poldi Pezzoli and painting signed: PETRVS PERVSINVS PINXIT, painted by PERUGINO, Pietro. Table 5 shows the results for the timeframe tasks. The most obvious result is that particular years are classified within each timeframe to describe each class, e.g., 1640 and 1635 are used to describe the timeframe of 1601–1650, while 1670 s is used to describe the timeframe of 1651–1700. The presented way of learning the representation seems to be a good way of learning domain-specific word embeddings. The embedding similarity could find good use in a specialized art search engine.

### 6.3. Interpreting the Classification Results

We used t-SNE to visualize the model output embeddings, including the classification tasks for type, school and timeframe in Figure 6. The first row shows the output of the ResNet50 model, while the second row shows the output of our ArtGCN models. Please note that both the paintings and artistic comments contain category information; and the category represented by the color corresponds to Figure 2. Overall, several observations are worth attention. In total, the accuracy score when using the ArtGCN model are higher than those when using ResNet50. However, for some specific categories (i.e., interior, landscape and still-life), ResNet50 performs better than does ArtGCN on the type classification task. For the school and timeframe classification tasks, ArtGCN outperforms ResNet50 on all the categories. Timeframes are more likely to be divided into adjacent periods rather than other, more distant periods. From experimental results, we can see that the proposed ArtGCN can achieve state-of-the-art classification results and learn painting and word embeddings. However, one of the main limitations of this study is that the GCN model is inherently transductive, test and validation artistic comment nodes(without labels) are included in GCN training. Thus, a major limitation of our study is that our method could not generate embeddings and make predictions for new paintings. Possible solutions to the problem are introducing inductive [52] or fast GCN model [53].

### 6.4. Painting Retrieval

In addition to exploring task-specific classification, we aimed to retrieve paintings of similar categories. We performed retrieval tasks for type and authors rather than for all 4 tasks because timeframe and school are highly related to the authors. To accomplish the retrieval goal, we used the various models to perform feature extraction and then calculated the similarity between feature vectors using cosine distance. Starry Night (Figure 7) is an oil-on-canvas painting by the Dutch post-impressionist painter Vincent van Gogh. In this painting, he vividly depicts a starry sky full of movement and change using exaggerated techniques. The entire picture is subsumed in a turbulent blue-green torrent. The rotating, restless and curling nebulae make the night sky seem extremely active. This unreal scene reflected Vincent van Gogh’s restless emotions and the illusory world of his madness. The Dream of Solomon (Figure 8) was painted by Luca Giordano; it depicts the story of Solomon from the Bible. During one night in a dream, Solomon asked the Lord to help him distinguish between right and wrong in his heart. The Lord was pleased that Solomon asked for this, and gave him a wise and discerning heart as well as wealth and honor. The Picnic (Figure 9) was painted by Monet, who used Impressionist light and shadow techniques to depict people relaxing on a grassy sward. The use of external light lends the picture a colorful, fresh and hearty effect. We used Reset50 (left) and ArtGCN (right) models to retrieve the paintings. The first row shows paintings retrieved by style, while the second row shows paintings retrieved by author. We have labeled the different retrieval categories below the paintings(for both style and author retrieval, we only tag the categories that differ). As we can see, all the styles are just same; only a small author category is different. Although these 2 models achieve great performance, the results are quite different. These results show that the retrieval mechanism of the two methods is completely different—even though both models can retrieve oil paintings correctly.

## 7. Conclusions

In this study, we used artistic comments to classify painting categories via a developed network named ArtGCN. We compared the results of ArtGCN with those of Resnet models, which use the actual image pixels to recognize painting categories. The experiments show that our method achieves significant increases in accuracy on 4 classification tasks. This result means that—in contrast to the painting itself—using other labeled information for paintings, such as artistic comments, also contains important information that can be used to train a computer to represent the art. In addition, we performed word embeddings tagged the labels using the embedding dimension with the highest value and extracted 10 words that have the highest values in the tagged dimension. We found that these higher-valued words are closely related to the categories. In other words, the extracted words can be used to better describe this category. We visualized and analyzed some phenomena worthy of attention. Both the methods based on computer vision and those artistic comments perform well on various painting classification tasks. We also constructed a painting retrieval system that can be used to enhance the capabilities of search systems in different online art collections. We tested the painting retrieval of the developed models using 3 paintings. The results of the two types of models are quite different and show that the art representation learned by computers is quite different when trained using a method based on computer vision versus a method based on artistic comments (natural language processing).

In this study, we used color information from the paintings and features from the artistic comments separately to address different painting classification tasks. In future work, we will attempt to combine these two methods by using the teacher model [54] and other methods. In addition, some labeling information exists that was not used in our experiments, and we believe there may be some inner relations among this additional information. For example, Vincent van Gogh’s paintings have a high probability of being landscape or portrait styles, while all the paintings of Giotto di Bondone are related to religion. Therefore, we will attempt to incorporate these additional features to build classification approaches based on both color features and other information. Lastly, an important indicator of how well the models perform would be a comparison with human performance on the task, i.e., the ability of experts and non-experts to guess the painting attributes just from the comment, we will also try to compare our methods with human performance.

## Figures and Tables

**Figure 1 sensors-21-01940-f001:**
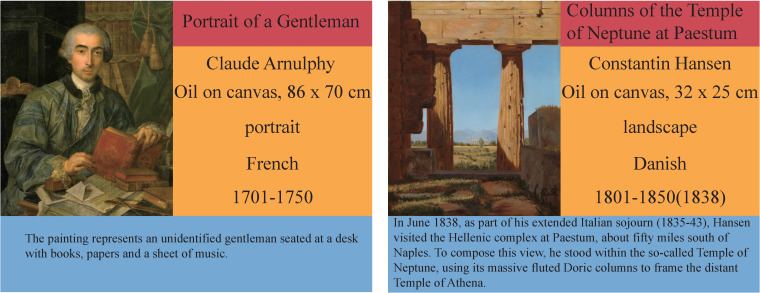
SemArt dataset. Each record includes a title (red), the author, technique and format, school, timeframe (yellow) and artistic comments (blue).

**Figure 2 sensors-21-01940-f002:**
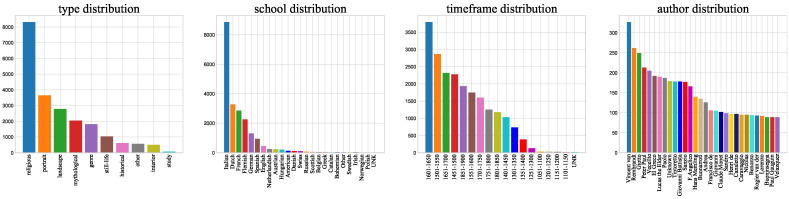
Class distribution of the style (**left 1**), school (**left 2**), timeframe (**right 2**), author (**right 1**). To better display the author distribution, we show only the top 30 artists; their names have been simplified to improve the figure.

**Figure 3 sensors-21-01940-f003:**
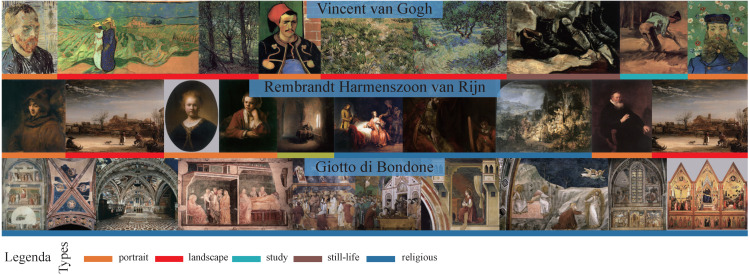
Paintings from the dataset adopted in this work. Each row contains samples from a different artist. The color coding represents different styles.

**Figure 4 sensors-21-01940-f004:**
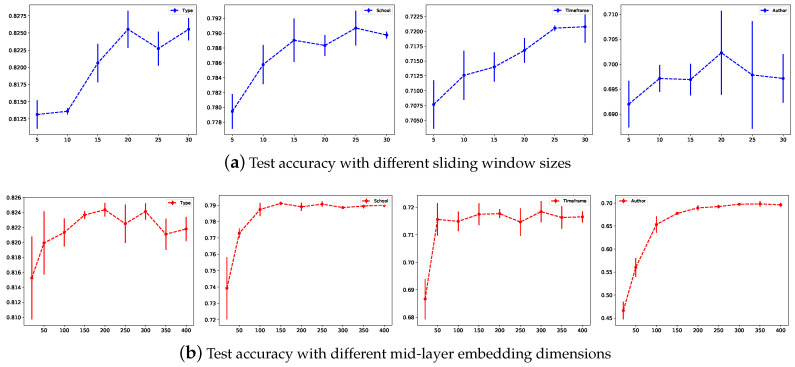
Test accuracy under different sliding window sizes and mid-layer embedding dimensions.

**Figure 5 sensors-21-01940-f005:**
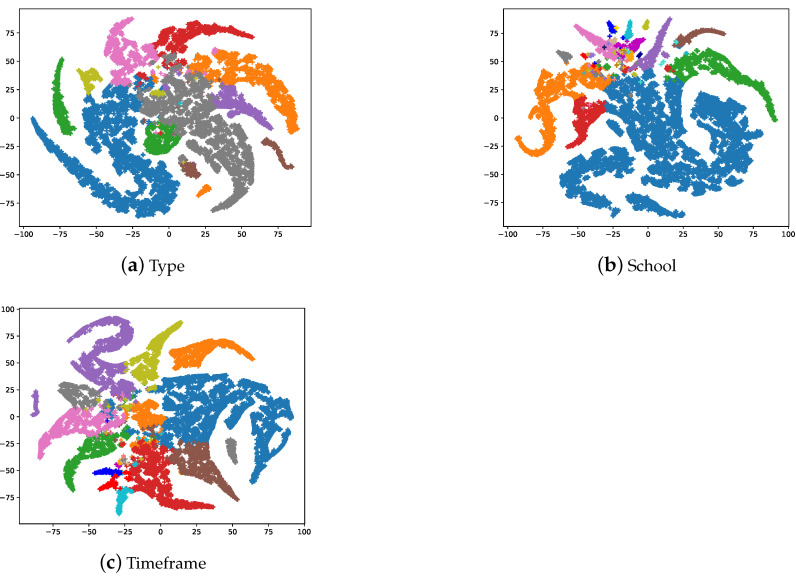
The t-SNE visualization of the second layer word embeddings. Each node represents the unique words in artistic comments. There are a total of 17,944 nodes (the number of words contained in artistic comments); we set the highest dimension as the word’s label.

**Figure 6 sensors-21-01940-f006:**
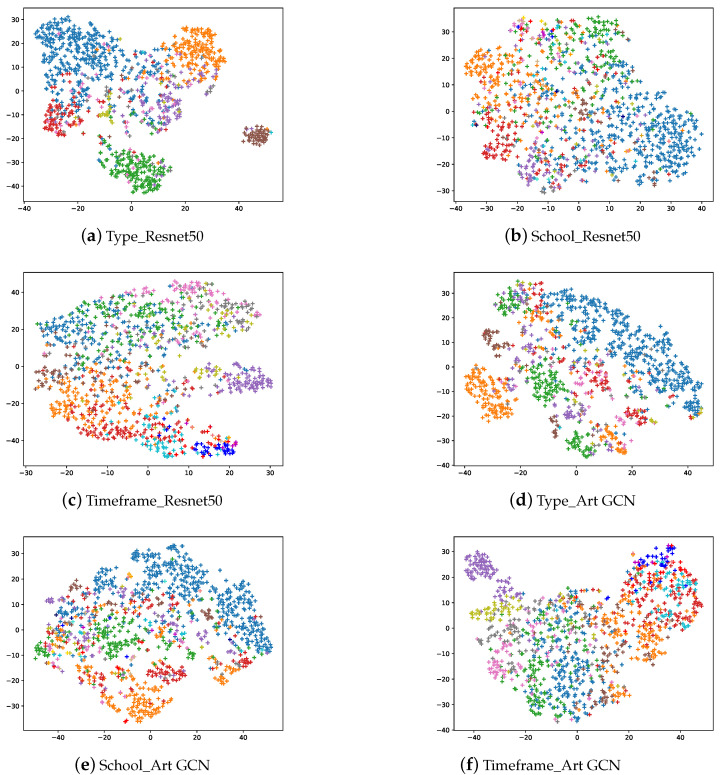
The t-SNE visualization of the image embedding and artistic comment embedding in the test set.

**Figure 7 sensors-21-01940-f007:**
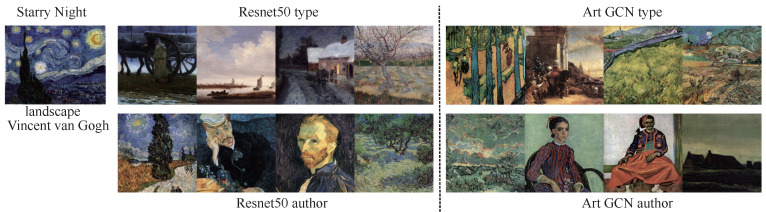
Similarity results for the painting Starry Night by Vincent Van Gogh.

**Figure 8 sensors-21-01940-f008:**
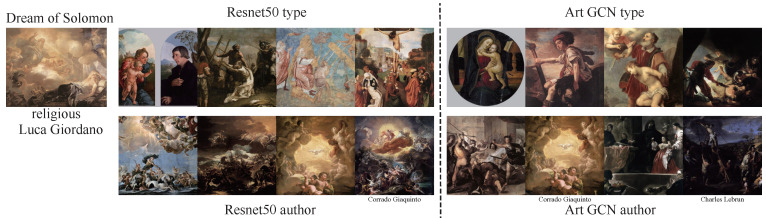
Similarity results for the painting Dream of Solomon, by Giordano, Luca.

**Figure 9 sensors-21-01940-f009:**
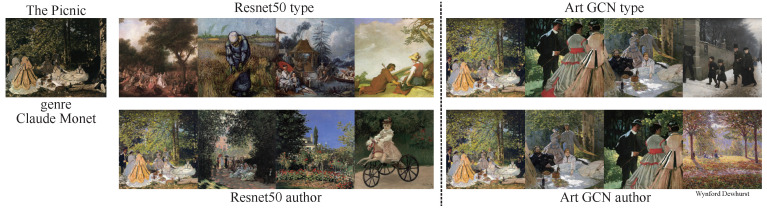
Similarity results for the painting The Picnic, by Monet, Claude.

**Table 1 sensors-21-01940-t001:** Dataset characteristics, including the number of categories in the different classification tasks based on the paintings in the training set.

Dataset	# Paintings	# Words	# Nodes	# Average Length	# Classes	
train	19,244	-	-	-	11	TYPE
val	1069	-	-	-	25	SCHOOL
test	1069	-	-	-	18	TIMEFRAME
total	21,382	17,944	39,326	59.27	350	AUTHOR

**Table 2 sensors-21-01940-t002:** Art classification results on SemArt Datasets.

# Region	# Model	# TYPE	# SCHOOL	# TIMEFRAME	# AUTHOR	# AVE
cv	resnet50 [47]	0.787	0.636	0.592	0.557	0.643
	resnet101 [47]	0.771	0.655	0.591	0.519	0.634
	resnet152 [47]	0.806	0.644	0.615	0.546	0.653
	mtl-resnet50	0.790	0.667	0.616	0.526	0.650
	kgm-resnet50 [48]	0.815	0.671	0.613	0.615	0.679
nlp	TF-IDF+LR	0.772	0.688	0.480	0.097	0.509
	fastText [49]	0.787	0.757	0.665	0.498	0.677
	fastText (bigrams)	0.804	0.774	0.634	0.453	0.666
	RoBERTa [50]	0.815	0.783	0.545	0.465	0.652
	mtl-ArtGCN	0.815	0.783	0.707	0.686	0.748
	ArtGCN	**0.826**	**0.788**	**0.717**	**0.702**	**0.758**

**Table 3 sensors-21-01940-t003:** Words with highest values for 8 classes using second-layer embedding. We show the top 10 words for each class. (Artists comments classified by style).

Religious	Portrait	Landscape	Mythological	Genre	Still-Life	Historical	Other
saints	portrait	estuary	bacchus	bambocciata	porcelain	battle	painting
triptych	sitter	ruisdael	scorpio	singerie	nots	alexander	,
mary	portraits	views	aquarius	ceruti	shrimps	fleet	’s
virgin	camus	coastal	capricorn	bamboccio	blackberries	brutus	artist
angels	portraitist	moored	pisces	steen	blooms	army	painted
madonna	hertel	boats	ovid	lhermitte	hazelnuts	war	one
altenburg	sitters	waterfalls	ariadne	singeries	tulips	defeated	\(
altarpiece	dihau	topographical	sagittarius	laer	grapes	naval	\)
polyptych	countess	hobbema	goddess	metsu	figs	king	painter
deposition	morbilli	fishing	pan	mieris	medlars	havana	paintings

**Table 4 sensors-21-01940-t004:** Words with highest values for several classes using second-layer construction. We show the top 10 words for each class. (Artists classified by school from comments).

Italian	Dutch	French	Flemish	German	Spanish	English	Netherlandish
chapels	leiden	fran	bruges	cranach	juan	british	bosch
pezzoli	nieuwe	ch	rubens	halle	zquez	sickert	bruegel
petrvs	rembrandt	fragonard	brueghel	herlin	vel	groom	haywain
petronio	bredius	boucher	memling	nuremberg	carlos	stubbs	hell
esther	kerk	le	snyders	holbein	alonso	maitland	geertgen
mariotti	hals	courbet	eyck	luther	vicente	starr	aertsen
pesaro	hague	ois	pourbus	heyday	las	wright	devils
evangelist	haarlem	bruyas	balen	secession	bautista	1st	bouts
florentine	hooch	lautrec	neeffs	liebermann	retablo	gainsborough	obverse
peruzzi	dutch	oudry	rubens’	friedrich	caj	sidney	sins

**Table 5 sensors-21-01940-t005:** Words with highest values for several classes using second-layer construction. We show top 10 words for each class. (Artists classified by timeframe from comments).

1601–1650	1501–1550	1651–1700	1451–1500	1851–1900	1551–1600	1701–1750	1751–1800
poussin	rer	1660s	ghirlandaio	brittany	arcimboldo	ricci	pulcinella
barberini	leo	vermeer	piero	parisian	zelotti	rosalba	reynolds
vel	capricorn	hooch	botticelli	ferenczy	sofonisba	ballroom	nemi
manfredi	scorpio	carre	mantegna	fattori	tintoretto	watteau	1773
caravaggism	aquarius	terborch	memling	poster	el	pellegrini	wright
1640	begat	1670s	bellini	cassatt	grandi	lancret	1777
ribera	1525	dou	roberti	boldini	zuccaro	tiepolo	volaire
hals	gossart	maes	tura	fantin	greco	crespi	1768
1635	raphael	steen	cossa	pouldu	veronese	carriera	zianigo
haarlem	sagittarius	deventer	signorelli	nabis	dell’albergo	boucher	gherardini

## Data Availability

Not applicable.

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
