# Peer review of "How to Represent Paintings: A Painting Classification Using Artistic Comments"

_sensors, 2021, doi:10.3390/s21061940_

Round 1
Reviewer 1 Report
Summary: The paper introduces a method for performing label classification on paintings based on their associated comments in the SemArt dataset. The proposed method builds a graph with the words and concepts in the comments and uses a 2-layer GCN to predict the author, timeframe, school and type labels. The method is evaluated and compared against vision-based methods on the SemArt dataset.
Comments:
- It is not clear how the method works at test time. The paper describes the graph construction and the GCN training process, but how are new samples in the test set classified? If we assume the test samples are not used in the training process and are not used to build the graph, some of the words in their comments would not be contained in the graph. If only the nodes that are in the graph are used, I assume different nodes may be assigned to different labels, so how is the final label decided? The test process should be clearily described in the paper.
- Some important details on the graph construction part are missing, e.g. which images are used to build the graph: training split only or all the splits?
- How are the test and validation accuracies computed?
- It is not surprising that accuracy is boost with respect to vision models as many of the painting comments may include the author and year information on them. To asses how big the impact of having the author information on the comment it would be necessary to: a) measure the % of comments that contain the ground truth label (author, year, etc.), b) run experiments and evaluations with the name of the author (and other ground truth labels) removed.
- Results presented in Table 2 are unfair. Vision-based and text-based methods cannot be compared directly, as the input is completely different. The table should at least include a clear way to state which methods are vision-based and which ones are text-based.
- The method only compares agains vision-based models. As there are no other text-based models evaluated on SemArt, the paper should at least present some text-based baselines to compare their method in a fair way. Some examples: a) TF-IDF + fully connected layer, b) average word2vec + fully connected layer, c) fine-tunned BERT, etc.
- The paper should be consistent with the names of the attributes. The type attribute is sometimes refered as "type" (L87), others as "style" (Figure 3), others as "class" (L91), etc.
- The title is a bit missleading "How to understand paintings: A painting classification using artistic comment". The proposed model does not do "understanding" or any reasoning of the paintings.
Missing references:
Text and art
- Garcia, Noa, et al. "A Dataset and Baselines for Visual Question Answering on Art." European Conference on Computer Vision. Springer, Cham, 2020.
- Sheng, Shurong, and Marie-Francine Moens. "Generating captions for images of ancient artworks." Proceedings of the 27th ACM International Conference on Multimedia. 2019.
- Baraldi, Lorenzo, et al. "Aligning text and document illustrations: towards visually explainable digital humanities." 2018 24th International Conference on Pattern Recognition (ICPR). IEEE, 2018.
Paintings classification
- Huckle, Nikolai, Noa Garcia, and Yuta Nakashima. "Demographic influences on contemporary art with unsupervised style embeddings." European Conference on Computer Vision. Springer, Cham, 2020.
- Chen, Liyi, and Jufeng Yang. "Recognizing the Style of Visual Arts via Adaptive Cross-layer Correlation." Proceedings of the 27th ACM International Conference on Multimedia. 2019.
- Wynen, Daan, Cordelia Schmid, and Julien Mairal. "Unsupervised Learning of Artistic Styles with Archetypal Style Analysis." NeurIPS 2018-Annual Conference on Neural Information Processing Systems. 2018.
Reviewer 2 Report
The paper presents an alternative way of classifying properties of paintings, namely its genre (called type in the paper), school, approximate year of origin, and author. Unlike methods presented in previous work, this classification is not based on the visual appearance of the painting, but on a short text description that is called "artist comment" and approached as a text classification problem. The authors chose a rather unusual representation for the comments. It is not based on the co-occurrence of words within a single text as in classical word embeddings (Word2Vec, GloVe) or contextual text representation (ELMo, BERT), but on word co-occurrence across various comments. The co-occurrence patterns are represented as a graph that is later processed by a graph convolutional network. Using this approach, the authors report significant classification improvement over the previous methods based on the visual appearance of the paintings.
Although the paper seems to present good quantitative results. I have serious doubts about the motivation of the work, its usefulness, and the rather unusual approach to text classification.
Concerns regarding motivation and usefulness
The paper lacks a deeper discussion on the motivation of using the method.
Classifying paintings based on their visual appearance seems like a pretty natural use case. From the practical side, when it would be good enough, it could assist an expert with describing a newly discovered piece of art. From a more theoretical side, it emulates what a knowledgeable person would think about a painting after seeing it. The research question is then: Can a machine emulate at least some part of human understanding of paintings?
The method presented in the paper, however, assumes that we have a short expert comment on a painting, but we do not know who the author is, what is the genre of the painting, when it was created, and what school the author belongs to. I can hardly imagine a situation when we have an expert comment on the paper, do not know this basic information. The person providing the comment knows the main attributes of the painting or at least is capable of making an educated guess.
If the motivation is not practical, then the motivation might be theoretical. Then the research question might be: Given an expert comment, can we estimate the basic data about a painting? It seems to me that the answer is almost certainly yes and answering the question experimentally does not seem like a big scientific contribution then.
The motivation discussed in lines 30-35 seems rather dubious to me. Language has a referential function, i.e., it is capable of referring to objects in the world and the object also has a visual representation, therefore language can (at least to some extent) describe visual perception. It has nothing to do with novels or with imagination. The model that you present does not attempt to emulate human visual imagination.
When using directly the visual data as the input, the models need to sort of overcome a bearer between the visual and textual modality. The paintings are visual artifacts whereas the tasks are defined using language categories. It is a difficult problem. When you use textual data, you are making the tasks easier because they do not have to do the hard step of crossing the modality bearer.
Concerns regarding the method
The paper lacks comparison with standard text-classification methods. Here, the appropriate baselines might be:
- a linear classifier with TF-IDF-based features,
- a linear classifier based on pre-trained word embeddings (I would recommend FastText),
- a classifier based on fine-tuning a pre-trained Transformer model (I would recommend RoBERTa).
In my opinion, the unusual design of the CGN would only be justifiable if the method would perform comparably with the SoTA text classification methods which I currently have serious doubts about.
Lastly, an important indicator of how well the models perform would be a comparison with human performance on the task, i.e., the ability of experts and non-experts to guess the painting attributes just from the comment.
Another weakness of the method that is not discussed in the paper is that it does not easily allow classifying new examples that were not part of the graph during training. If I understand the graph construction correctly (and especially Equation 9), this is a fundamental property of the graph because there are oriented edges leading from the comment nodes to the lexical nodes. However, having the edges oriented in the other direction would fix this issue.
Furthermore, the method is presented in a rather clumsy way and the description is difficult to follow (see the next section of the review).
Not to be only critical, the presented way of learning the representation seems to be a good way of learning domain-specific embeddings (Tables 3-5). The embedding similarity could find good use in an art search engine specialized.
Detailed comments on the text
- line 10: You report the average accuracy gain in the abstract, but at this point, it is not clear, how comparable the tasks are. If the classification tasks differ in difficulty, it doe not make sense to report the average.
- line 52: I would avoid using the term of understanding art. It is a totally vague term. I am not able to provide a good definition of what it is, but I am sure that what the models do is far from everything we might even consider to be understanding. You can say that you compare two ways of representing painting using neural networks.
- Section 2.2: Since 2018, Transformers are the state-of-the-art architecture for most NLP tasks. The current SoTA approach to basically text classification is taking a pre-trained Transformer model (such as BERT) and finetune it for a particular task.
- line 91: You say "including" which implies that you name some examples, but what follows is a complete list.
- line 100: Typesetting the artists' names like this looks strange. I would either typeset them as in a normal text: Vincent van Gogh or put the names into parentheses.
- Figures 2 and 4: labels are too small.
- Section 4.2 where did line numbering disappear?
- §4.2, preprocessing: why removing tokens with less than five characters. Many important keywords can be shorter.
- Algorithm 1: This is a very standard training loop, there is no need to put it explicitly in the paper in this form.
The method description is not well structured. The crucial idea here is how you construct the graph and what the graph is. Before figuring it out, the readers need to go through details that do not make much sense before knowing how do you construct the graph and when the readers figure it out, they have to return to the details to understand the method fully.
You should start with saying that there are nodes of two types:
- Nodes representing artist comments represented as TF-IDF weighted bag of words,
- Nodes that correspond to unique words.
Then, you can say what the edges are. In this way, you got the first layer of your GCN and then you finally say that you iteratively apply Eq. 1.
Round 2
Reviewer 1 Report
I would like to thank the authors for the efforts made to include my suggestions on the manuscript. I think most of the unclear parts of the paper have been now improved. I also aprreciate the clarification on Line 275 about the limitations of using a transfuctive approach.
Some minor comments:
* Figure 2: increase font size
* Table 1: It looks incomplete. Better to provide info for each of the rows
* Line 174: exact --> extract
* Table 2: add a line to separate cv and nlp methods
* Table 2: add references to the original methods
* Table 2: mtl-resnet50 has not been described in Section 5.2
* Table 2: Add results of paper [54]
Reviewer 2 Report
Since the previous submission, the paper got substantially improved, including the description of the architecture which is a central point for understanding the paper. Especially given the fact that the method outperforms standard text classification methods, I think the paper is worth publishing, although I still have some doubts about the motivation of the work.
Some minor notes:
- L. 205: The description of what you did with RoBERTa does not make sense. I guess you mean that you finetune it the same as it is done in the scripts for the GLUE benchmark in the Huggingface implementation. (Otherwise, it would not make any sense to finetune it for the GLUE tasks that do not have anything to do with your classification tasks).
- You can also tabulate the number of parameters of the individual models. Your model is probably much smaller than RoBERTa in terms of parameters and still performs better. This is a good selling point, too.
- Figure 5: It is not clear what the colors code. I guess these are different target classes. Also, what does `_word` mean in the captions?
- The confusions matrices in Figure 7 are not readable at all. At the first sight, it does not seem to me that they would anything really interesting (such as some frequent misclassification). If my impression is correct and if you do not find a way how to make them more readable, you can probably drop them.
